# Neuroendovascular Surgery Applications in Craniocervical Trauma

**DOI:** 10.3390/biomedicines11092409

**Published:** 2023-08-28

**Authors:** Michael Kim, Galadu Subah, Jared Cooper, Michael Fortunato, Bridget Nolan, Christian Bowers, Kartik Prabhakaran, Rolla Nuoman, Krishna Amuluru, Sauson Soldozy, Alvin S. Das, Robert W. Regenhardt, Saef Izzy, Chirag Gandhi, Fawaz Al-Mufti

**Affiliations:** 1Department of Neurosurgery, Westchester Medical Center at New York Medical College, Valhalla, NY 10595, USA; 2Department of Neurology, Westchester Medical Center at New York Medical College, Valhalla, NY 10595, USA; 3Department of Neurosurgery, University of New Mexico, Albuquerque, NM 87108, USA; 4Department of Surgery, Division of Trauma and Acute Care Surgery, Westchester Medical Center at New York Medical College, Valhalla, NY 10595, USA; 5Department of Neurology, Maria Fareri Children’s Hospital, Westchester Medical Center at New York Medical College, Valhalla, NY 10595, USA; 6Goodman Campbell Brain and Spine, Indianapolis, IN 46032, USA; 7Department of Neurology, Beth Israel Deaconess Medical Center, Harvard Medical School, Boston, MA 02215, USA; 8Department of Neurology, Massachusetts General Hospital, Harvard Medical School, Boston, MA 02215, USA; 9Department of Neurology, Brigham and Women’s Hospital, Harvard Medical School, Boston, MA 02215, USA

**Keywords:** neuroendovascular surgery, cerebrovascular injuries, neurointerventional, head and neck trauma, craniocervical trauma, angiography

## Abstract

Cerebrovascular injuries resulting from blunt or penetrating trauma to the head and neck often lead to local hemorrhage and stroke. These injuries present with a wide range of manifestations, including carotid or vertebral artery dissection, pseudoaneurysm, occlusion, transection, arteriovenous fistula, carotid-cavernous fistula, epistaxis, venous sinus thrombosis, and subdural hematoma. A selective review of the literature from 1989 to 2023 was conducted to explore various neuroendovascular surgical techniques for craniocervical trauma. A PubMed search was performed using these terms: endovascular, trauma, dissection, blunt cerebrovascular injury, pseudoaneurysm, occlusion, transection, vasospasm, carotid-cavernous fistula, arteriovenous fistula, epistaxis, cerebral venous sinus thrombosis, subdural hematoma, and middle meningeal artery embolization. An increasing array of neuroendovascular procedures are currently available to treat these traumatic injuries. Coils, liquid embolics (onyx or n-butyl cyanoacrylate), and polyvinyl alcohol particles can be used to embolize lesions, while stents, mechanical thrombectomy employing stent-retrievers or aspiration catheters, and balloon occlusion tests and super selective angiography offer additional treatment options based on the specific case. Neuroendovascular techniques prove valuable when surgical options are limited, although comparative data with surgical techniques in trauma cases is limited. Further research is needed to assess the efficacy and outcomes associated with these interventions.

## 1. Introduction

Traumatic injuries to the head and neck can result in adverse neurological outcomes as they involve critical neurovascular anatomy [1]. Occurring in 3–20% of patients following craniocervical trauma, traumatic cerebrovascular injury is a frequent cause of morbidity and mortality [2]. Although surgical and medical approaches have historically guided the management of these serious and potentially fatal injuries, endovascular techniques have emerged due to their minimally invasive nature. Although once limited to the treatment of traumatic injuries of the carotid and vertebral arteries after conservative management has failed, advancements in technology, coupled with improvements in the safety profile and reduced complication rates, have allowed the field to expand greatly over the past several decades.

Injuries such as blunt cerebrovascular injury (BCVI), dissection, pseudoaneurysm, vessel occlusion, carotid-cavernous fistula (CCF), arteriovenous fistula (AVF), cerebral venous sinus thrombosis (CVST), epistaxis, and subdural hematomas (SDH) can be difficult to diagnose, particularly in polytrauma patients with impaired neurological functioning. Furthermore, many of these injuries can develop over time and may not be present in the initial evaluation. Fortunately, the implementation of comprehensive screening protocols has led to increased detection of these injuries. Currently, the preferred management algorithm of choice for stable patients involves initial screening with noninvasive imaging, such as computed tomography angiography (CTA) and magnetic resonance angiography (MRA), followed by subsequent selective use of digital subtraction angiography (DSA) [3]. DSA remains the gold standard for diagnosing traumatic carotid and vertebral artery injury as it provides superior visualization of both arterial anatomy and real-time blood flow, allowing the physician to visualize both obvious and subtle flow-altering injuries. Critically, these injuries can also be treated during the same procedure, providing both diagnostic and therapeutic benefits. Additionally, the physician can examine the collateral circulation in the region supplied by traumatized vessels, which is crucial in determining the optimal treatment modality.

Endovascular techniques, such as coiling, thrombectomy, stent placement, and embolization, have evolved from being merely an adjunct to surgical treatment to a now viable alternative for initial management. These minimally invasive techniques may help reduce the morbidity and mortality rates involved in the treatment of these devastating injuries [1,4,5,6]. While a promising approach for treating various cerebrovascular disorders, the potential risks and limitations of these evolving techniques must be considered. Inherent to any endovascular procedure is the possibility of complications, such as vessel perforation, embolization, or hemorrhage, which can lead to neurological deficits or even fatal outcomes. Moreover, long-term durability remains a concern, as some treated aneurysms or arteriovenous malformations may recur or develop new lesions over time, requiring subsequent interventions [7,8,9]. Although these advancements show great potential, establishing a consensus on the diagnosis and management of traumatic head and neck injuries requires further investigation through randomized clinical trials comparing surgical and endovascular treatments [1,10,11]. In this paper, we provide an updated and comprehensive nonsystematic review of the role of endovascular treatments in the management of traumatic cerebrovascular injuries.

## 2. Methods

The authors conducted a PubMed search spanning 1989–2023, including the terms: endovascular, trauma, dissection, blunt cerebrovascular injury, pseudoaneurysm, occlusion, transection, vasospasm, carotid-cavernous fistula, arteriovenous fistula, epistaxis, cerebral venous sinus thrombosis, subdural hematoma, and middle meningeal artery embolization.

## 3. Blunt Cerebrovascular Injury

Blunt Cerebrovascular Injuries (BCVI) result from high-energy nonpenetrating blunt force trauma or high-speed deceleration injury to the neck and typically involve either the internal carotid artery, vertebral artery, or both. Most frequently due to motor vehicle accidents, BCVI occurs in approximately 1–2% of blunt trauma patients and in 2.4% of trauma patients that require inpatient care over 24 h [10,11,12]. BCVI is often categorized into five grades according to the scheme proposed by Biffl et al. [13,14]. Grade I involves intimal irregularity with <25% vessel stenosis; grade II denotes a dissection with >25% vessel stenosis; grade III signifies the formation of a pseudoaneurysm; grade IV entails vessel occlusion; and grade V indicates transection of the vessel with active extravasation (Figure 1).

Historically, BCVI was not diagnosed until the onset of stroke due to inaccessibility to noninvasive imaging. However, the implementation of comprehensive screening protocols has resulted in the earlier initiation of appropriate therapies in BCVI patients, in addition to significantly increasing the detection of these injuries [1]. The detection of these injuries, classically underdiagnosed, has significantly risen from 0.1% to 3% of polytrauma patients due to advances in noninvasive imaging, such as CTA and MRA, as well as the development of improved screening protocols [15,16,17]. Offering similar detection rates to DSA, a multi-slice CTA with an eight or greater multidetector has been recommended as an alternative screening method by both the Western Trauma Association (WTA) and the Eastern Association for the Surgery of Trauma (EAST) in their BCVI guidelines [18,19]. Hundersmarck et al. recently found that the incidence of BCVI has increased with improved CTA diagnostic modalities [20]. However, a recent meta-analysis investigating the diagnostic accuracy of CTA to DSA for detecting BCVI in trauma patients revealed moderate to good specificity but low sensitivity for CTA compared to DSA. The authors also noted that an increase in channels to 16–64 slices did not improve diagnostic accuracy compared to fewer (<16) slices [21].

Mechanistically, BCVI is the result of longitudinal stretching and injury to the vessels as the head and neck are forcefully moved in flexion, extension, or rotation. Rapid acceleration–deceleration (e.g., motor vehicle crash) can cause rotation and hyperextension of the neck, stressing the craniocervical vessels [1,11,14]. This can cause an intimal tear with exposure of subintimal layers to the blood flow and, consequently, thrombus formation, wall hematoma, and even lumen occlusion [1,19]. In some instances, this process develops into a pseudoaneurysm [6,7].

The signs and symptoms of BCVI may not always be obvious, especially in complex trauma patients with limited neurological function. BCVI carries a significant risk of morbidity and mortality, with rates of 48–56% and 23–30%, respectively [12,18]. One case series found that patients with traumatic vertebral artery dissections had higher rates of in-hospital stroke and longer lengths of intensive care unit and hospital stay. A series of 14 studies performed by Scott et al. at a level one trauma center from 2003 to 2013 revealed a 1–1.7% stroke rate in grade I and II BCVI and 7% in grade III and IV BCVI [22,23,24,25]. More recently, a meta-analysis of treatment for asymptomatic BVCI involving 19 studies found that any medical treatment—including aspirin, Plavix, warfarin, and heparin—was better than no treatment, but the authors did not find any difference in outcomes among treatment options chosen [26]. Management options and outcomes for BCVI are summarized in Table 1.

### 3.1. Grade I–II: Dissection

Neuroendovascular techniques have limited utility in the treatment of grade I and II BCVI, where the mainstay of treatment is an antithrombotic agent, such as aspirin or heparin. Endovascular treatments are employed only when these low-grade injuries progress to higher grades [18,19]. According to the updated 2020 EAST BCVI guidelines, the previously routine placement of stents as an adjunct to antithrombotic therapy is not recommended for adult patients with grade II BCVI.

In a study conducted by Scott et al. (2015), which included a cohort of 100 patients with 117-grade I and II carotid artery BCVIs, it was found that 19 patients had worsening injuries on follow-up imaging [22]. Among these 19 patients, 17 developed pseudoaneurysms, and three of them required endovascular stent placement due to radiographic progression. Similarly, Scott et al. (2014) examined a series of 120 patients with 152-grade I and II vertebral artery BCVI. Of these, they identified nine patients with worsening injuries, with seven developing pseudoaneurysms. However, none of these patients required endovascular treatment [23].

### 3.2. Grade III: Pseudoaneurysm

Traumatic pseudoaneurysms arise from a disruption of the internal elastic lamina with subsequent involvement of all layers of the arterial wall. Expansion of the resulting false lumen within the arterial wall can compress the true lumen of the artery, resulting in stenosis and possible ischemic or embolic infarcts. Furthermore, this pseudoaneurysm has the potential to rupture, causing extravasation into the surrounding tissue. Therefore, patients can present with a variety of symptoms, including Horner syndrome, neck pain, pulsatile neck or scalp mass, epistaxis, dysphonia, dysphagia, upper airway compromise, hemiparesis, or even coma [5,6].

While both the WTA and EAST guidelines recommend screening using a CTA, their treatment guidelines differ. Both state that initial treatment should be with an antiplatelet agent or anticoagulation; however, the timing of antiplatelet therapy or anticoagulation is frequently complicated and delayed due to the polytraumatic injuries commonly associated with the high-force mechanism of injury required to produce grade III BCVI. In patients with severe luminal narrowing or expanding pseudoaneurysms, the WTA guidelines state that stent placement should be considered after a repeat CTA 7–10 days after the initial diagnosis is obtained [18]. On the other hand, the EAST guidelines now state that routine stent placement should not be performed as an adjunct to antithrombotic therapy in grade III injuries. However, they acknowledge that there may be select cases, such as an enlarging pseudoaneurysm, in which stent placement would be appropriate and potentially beneficial [17]. Neither set of guidelines provides specific recommendations on the preferred treatment modality, complicating matters for both patients and providers.

Given the lack of guidelines on the optimal technique and the complexity of polytrauma patients, treatment plans must consider both the associated injuries and specific patient factors. Complete occlusion of the pseudoaneurysm with preservation of the parent vessel is often possible, and good results have been obtained using coil embolization, coil embolization with stent placement, stent angioplasty, and stent placement with covered stents or flow diverters [6,7,27,28,29,30,31]. However, caution must be taken because stent placement requires dual antiplatelet therapy, and the risk of bleeding associated with dual antiplatelet agents must be weighed against the risk of complications from untreated pseudoaneurysms. In cases where therapeutic occlusion of the parent vessel is necessary, a balloon test occlusion (BTO) can be performed under local anesthesia to assess collateral flow both clinically and angiographically prior to therapeutic occlusion of the vessel.

Examining a series of 23 patients with grade III vertebral artery injuries, Scott et al. (2015) found that nine patients had stable BCVI, ten resolved, three improved, and one worsened. All except one were treated with antiplatelet or anticoagulation therapy. While patients were observed developing infarcts that were believed to be due to other concurrent BCVIs in the anterior circulation, none of the patients required endovascular therapy [25]. In another series of 44 patients with 53 grade III carotid artery BCVIs, eight patients required endovascular treatment, seven with stents, and one with coil embolization. Final follow-up imaging showed that six cases were resolved, 28 were stable as a result of treatment, and 12 worsened, prompting treatment in five patients. Three patients developed infarcts but did not require endovascular treatment [24]. The authors concluded that the post-traumatic infarction rate after high-grade BCVI may be overestimated in the literature.

### 3.3. Grade IV: Occlusion

The carotid and vertebral arteries may become occluded after traumatic injury, primarily due to vessel dissection or secondary to extrinsic compression from cervical spinal fractures. Occlusions caused by extrinsic compression rarely require neuroendovascular techniques but may require therapeutic occlusion of the proximal vessel if reduction of the fracture will result in further injury to the vessel [5]. Traumatic occlusions caused by dissections are treated based on the presence of ischemic stroke and adequacy of the collateral circulation.

The current WTA and EAST guidelines recommend that all patients with carotid or vertebral artery occlusions should be treated with antiplatelet or anticoagulation therapy to prevent further propagation of the intraluminal thrombus and potential embolic events. Furthermore, for patients with early neurologic deficits and accessible lesions who have not suffered a complete cerebral infarct, operative or interventional repair should be considered to restore flow [18,19]. Endovascular options include mechanical thrombolysis, mechanical thrombectomy, or stent angioplasty [5,32].

In a study of 42 patients with grade IV vertebral artery BCVIs, Scott et al. (2015) reported that at the latest follow-up, twenty-eight had stable occlusions, thirteen improved with asymptomatic recanalization, and two resolved completely [25]. Although 11 patients underwent coil embolization, the authors stated that these procedures were conducted early on, and the practice was subsequently discontinued. Three patients, one of whom had bilateral vertebral artery occlusions, had infarcts believed to be due to the BCVI, resulting in a 100% mortality rate. All but one were treated with antiplatelet or anticoagulation therapy.

In another series of eight patients with eight grade IV carotid artery BCVIs, Scott et al. (2015) reported that three died during or shortly after presentation, three had stable occlusions, and two improved with vessel recanalization. The five surviving patients were treated medically, with one experiencing an asymptomatic stroke. The authors concluded that all strokes caused by grade IV BCVI were likely present upon hospital admission. Furthermore, they suggested the need for revised follow-up and treatment protocols since there were no delayed infarcts in the post-injury period, and most occlusions either remained stable or improved on follow-up.

### 3.4. Grade V: Transection

Traumatic transection of the extracranial carotid or vertebral arteries with active extravasation is associated with high rates of morbidity and mortality, mandating immediate attempts to control bleeding due to the potential for significant blood loss [5]. The current WTA guidelines recommend urgent surgical repair for accessible lesions and endovascular techniques for inaccessible injuries [18]. However, these guidelines do not provide specific recommendations on which technique should be employed.

Injuries to the carotid artery in the neck (i.e., zone 2) are usually surgically accessible and can be readily treated. Conversely, transections of the carotid artery in zones 1 and 3, as well as vertebral artery transections, are more difficult to access due to the extensive exposure needed for proximal and distal control. These injuries are thus more amenable to endovascular treatments [5]. Other treatment options include therapeutic occlusion of the vessel proximal to the tear with or without distal occlusion using coils or liquid embolization materials. In such cases, temporary balloon occlusion testing can be performed prior to therapeutic occlusion. Reconstruction of the injured vessel can then be performed by stent placement. However, the requirement of antiplatelet medications for stent placement necessitates careful consideration of bleeding complications or progression of hemorrhage, particularly in the setting of active extravasation.

## 4. Intracranial Dissection

Intracranial dissections represent an uncommon and likely underdiagnosed phenomenon because of the inherent difficulty in visualizing the microscopic radiographic signs in pathologic intracranial arteries [33]. The absence of major randomized control trials has resulted in a lack of widely accepted management guidelines, particularly because patients can present either with ischemia or hemorrhage. Intracranial vessels, which lack an external elastic lamina and have minimal adventitial tissue, are prone to subadventitial dissections and concomitant subarachnoid hemorrhage [33,34,35].

Several small case series of intracranial dissections caused by all etiologies have shown that anterior circulation dissections are present in approximately 72.7–88% of cases with ischemia and 20–65% with subarachnoid hemorrhage, while posterior circulation dissections are seen in approximately 26–62% of ischemic cases and up to 70% of cases with subarachnoid hemorrhage [36]. Most intracranial dissections occur spontaneously, with only a small percentage attributed to trauma. Among 61 patients with intracranial dissections in a recent report, only 11.5% were due to trauma, and all of these were in the ICAs [37]. In the pediatric population, intracranial dissections most often involve the anterior circulation [38].

There is also an increased prevalence of spontaneous intracranial dissection among patients with certain connective tissue disorders, including Ehlers–Danlos syndrome, Marfan syndrome, Loeys–Dietz syndrome, and neurofibromatosis type I [39]. This is thought to arise due to mutations in the extracellular matrix proteins, such as collagen and proteoglycans, which cause vessel wall weakening, which predisposes to rupture. However, the exact prevalence of this association is unknown due to a dearth of case–control studies involving these patient populations. An association between intracranial dissection and fibromuscular dysplasia has also been recognized, but the ARCADIA-POL study (2019) was the first to quantify this, with 39.5% of patients in the study developing spontaneous cervical artery dissection [40].

The diagnosis of these injuries can be difficult due to a wide range of clinical presentations, varying from mild findings, such as facial hypoesthesia or dysmetria, to more severe deficits like aphasia, hemiplegia, and coma [5]. Furthermore, the majority of patients are asymptomatic in the acute phase. The optimal treatment approach for intracranial dissections is currently unknown, as the presenting radiographic findings can vary from segmental stenosis and occlusion to fusiform or saccular aneurysmal dilatation [35]. Therefore, the management of traumatic intracranial dissection is frequently individualized based on factors such as neurologic function, the presence of collateral flow, and other traumatic injuries. For example, dissections with non-flow-limiting stenosis or occlusions with adequate collateral circulation can often be managed medically, while those with flow-limiting stenosis and inadequate collateral circulation may be candidates for endovascular therapy [5]. Although asymptomatic patients with stable dissections are typically treated with anticoagulation or antiplatelet agents, medical therapy alone should be used cautiously due to the risk of subarachnoid hemorrhage [5,33].

In patients presenting with ischemia, endovascular therapy is usually reserved for those with worsening symptoms or progression of the dissection despite optimal medical management [33,35]. Reconstruction of the vessel can be performed using flow diversion or stents to separate the false and true lumens. This method leads to a controlled thrombosis of the dissection while preserving vessel patency and physiologic cerebral blood flow [38]. Overlapping stents and flow diversion without coil embolization have also been shown to be a safe option for dissecting aneurysms, as they avoid introducing coils into the thin-walled aneurysm, thereby reducing the risk of rupture [33]. Deconstructive techniques involve vessel takedown with embolization material or detachable coils. Additionally, emergent revascularization by mechanical or chemical thrombectomy can be considered in acutely symptomatic patients, similar to acute ischemic stroke therapy. This maneuver can be performed by itself or in conjunction with the aforementioned techniques [33].

Since up to 40% of patients rebleed within the first few days, with a mortality rate of up to 50%, patients presenting with subarachnoid hemorrhage from an intracranial dissection require more aggressive treatment than those presenting with ischemia [33,34,35]. Endovascular treatment options, including the techniques mentioned previously, are tailored to each patient based on vessel morphology, with the goal of reducing blood flow to the dissected region. While deconstructive techniques result in both rebleeding and infarction rates of <33%, reconstructive techniques have reported rebleeding rates of <50% with infarction rates of <14% [33]. Other novel approaches, such as the use of onyx (Medtronic, Minneapolis, MN, USA), a liquid embolic system composed of an ethylene vinyl alcohol copolymer to occlude a dissection and its associated pseudoaneurysm, have been reported [41]. It is also important to note that, in patients with previously diagnosed connective tissue disorders, the risks and benefits of surgical, endovascular, or conservative treatment are unknown at this time, and further studies are urgently needed in this area.

## 5. Post-Traumatic Vasospasm

Secondary injury is a cause of substantial morbidity and mortality following traumatic brain injury (TBI), and its impact can be exacerbated by hypotension and hypoxia. Recently, arterial vasospasm has come into focus as a potential contributor to secondary injury via delayed ischemia (Figure 2). Post-traumatic vasospasm (PTV) is hypothesized to result from the release of spasmogenic and neuroinflammatory substances generated by the breakdown of blood products in the subarachnoid space following trauma [42]. In contrast to aneurysmal subarachnoid hemorrhage, surveillance for PTV is not routinely performed, making it difficult to determine its true incidence. As imaging technology has improved, rates of PTV detection in TBI patients have been reported to range from 27% to 63% [43,44]. Diagnosis of PTV is often established using CTA/CT perfusion and DSA. Additionally, by monitoring the mean blood velocity, which acts as a surrogate for cerebral perfusion pressure, transcranial Doppler ultrasonography has been instrumental in guiding the early management of TBI patients [45]. Certain clinical features, such as fever, low Glasgow Coma Scale score, number of cerebral lobes affected by the traumatic injury, high Injury Severity Score, and presence of an associated pseudoaneurysm or hemorrhage, have been shown to correlate with an increased risk of developing PTV [43,46,47,48,49,50].

Currently, nimodipine, a calcium channel blocker (CCB), is the most effective and widely used medication for the prevention of PTV; however, interventional techniques, such as balloon angioplasty and intra-arterial vasodilator administration, are also used [51,52]. As is the case with many neurovascular injuries, no consensus guidelines exist on which patients should be screened for PTV, and treatment of this clinical phenomenon is thus not well documented. However, a recent meta-analysis of 14 studies involving 1885 PTV patients evaluated two patient groups: those who received a tailored therapeutic intervention with either treatment with CCBs, endovascular intervention, or dopamine-induced hypertension (n = 982), and those who did not receive any intervention. The researchers found that, of the 982 patients who received tailored intervention, the rate of favorable outcome was 94.1% for endovascular intervention (16/17 patients), 57.3% for CCBs (500/872 patients), and 54.8% for dopamine-induced hypertension (51/93 patients). However, the comparatively small sample size of the endovascular intervention group precludes any strong conclusions from being drawn [53]. Thus, further studies should be performed to best determine the optimal management for these patients, especially given the wide spectrum of clinical sequelae associated with vasospasm.

## 6. Carotid-Cavernous Fistula

Carotid-cavernous fistula (CCF), a type of arteriovenous malformation, is typically associated with complex fractures of the skull base and can form as a result of cranial trauma (Figure 3). Following injury, arterialization of the venous outflow from the cavernous sinus may result in the development of a myriad of symptoms, including visual deterioration, ophthalmoplegia, diplopia, headache, conjunctival chemosis, proptosis, pulsatile tinnitus, and ocular bruit [5,54,55,56,57]. These signs and symptoms may develop days to weeks following the initial traumatic event [5]. Initial evaluation of a CCF patient includes standard ocular tonometry, pneumotonometry, ultrasonography, color Doppler imaging, CTA, and/or MRA [56].

Since CCF can present with a variety of symptoms, a multidisciplinary approach involving experienced radiologists, ophthalmologists, and neurologists is essential to achieve a comprehensive assessment and appropriate management of carotid-cavernous fistula patients. Despite the advantages of these diagnostic tools, challenges may arise, such as difficulty in detecting small or low-flow fistulas with color Doppler imaging and potential variations in results due to operator-dependence [5,54]. Additionally, patient cooperation during the imaging procedure is also crucial, particularly in cases where severe pain or discomfort may affect image quality and diagnostic accuracy. Nonetheless, if suggested by noninvasive modalities, the diagnosis of a CCF is confirmed by DSA. Critical to its status as the diagnostic gold standard for these cerebrovascular injuries, DSA allows for the classification of CCFs, as well as its delineation of the arterial supply and identification of the fistulous point aids in treatment planning [55,58].

CCFs are classified as direct, in which there is a direct connection between the ICA and cavernous sinus, or indirect (dural), in which there are communications between the cavernous sinus and meningeal arterial branches [5]. Direct fistulas account for greater than 70% of cases and have a high flow rate, rapidly develop venous congestion and symptoms, and are often associated with trauma [54]. Indirect fistulas are low-flow, and their etiology is often unknown [5]. Because of their low flow status, indirect CCFs may be managed conservatively by observation, with spontaneous closure seen in as much as 70% of cases [59,60,61]. Medical therapy with intraocular pressure-lowering agents, intermittent compression of the ipsilateral ICA or superior ophthalmic vein, or radiosurgery may be considered adjuvant therapy, but these are rarely used [56].

Almost all CCFs can be treated using endovascular methods alone or in combination, including transarterial embolization, transvenous embolization, or therapeutic occlusion of the vessel. Indications for intervention include refractory elevation in intraocular pressure, severe diplopia, proptosis, optic neuropathy, retinal ischemia, severe ocular bruit, and cortical venous drainage. Cure rates approach 100% with endovascular techniques, with low rates of complications and mortality and the potential to restore oculomotor function and vision if early treatment is pursued [56,57,62]. Embolization may be achieved using detachable coils, balloons, liquid embolic agents, and/or flow-diverting stents, with the goal of therapy to occlude the fistulous point. Transarterial embolization is the preferred approach; however, if the fistulous point cannot be accessed via arterial catheterization, a transvenous approach may be attempted by utilizing the inferior petrosal sinus or superior ophthalmic vein [57]. For cases with ICA dissections, flow-diverting stents serve as a potential adjuvant therapy to coil embolization by way of endoluminal reconstruction [56,63,64]. However, the need for dual antiplatelet therapy is a disadvantage of stent placement. As a last resort, therapeutic occlusion of the parent vessel may be considered if the patient has adequate collateral flow [54]. A recent meta-analysis of 1494 CCF patients demonstrated that coiling, onyx, and balloons were the most common endovascular treatment modalities, and a high percentage of these patients experienced complete remission with improvement in clinical symptoms [65]. The most common clinical manifestations of CCF included proptosis, exophthalmos, chemosis, diplopia, and cranial nerve palsies.

## 7. Other Intracranial Arteriovenous Fistulas

Intracranial AVFs other than CCFs, although rare, can result from trauma to the meningeal branches of the external carotid artery (ECA) or as the sequelae of progressive stenosis or occlusion of a dural venous sinus [5]. As the pressure in the venous sinus rises, meningeal arteries form fistulous connections with dural sinuses or cortical veins. The development of an AVF can be a dynamic process, involving the recanalization of the thrombosed sinus and the recruitment of additional ECA feeders [66]. The resultant venous hypertension can lead to vasogenic edema, venous infarction, and ultimately intracerebral hemorrhage. Patients can present in a delayed fashion with a variety of symptoms as they are dependent on venous congestion and mass effect from hemorrhage rather than the specific location of the fistula. While CTA and MRA can screen for the presence of an AVF, DSA remains the gold standard for assessing the architecture of the AVF and locating the fistulous point.

Patients with AVF and cortical venous drainage who present with severe neurologic deficits have a poor prognosis, with annual rates of intracerebral hemorrhage and neurologic deficits ranging from 7.4% to 19% and an annual mortality rate of 3.8%. Urgent treatment is recommended for these patients to mitigate these risks. Asymptomatic patients with AVF and cortical venous drainage have a more benign course that nevertheless is still significant. Their annual rates of hemorrhage and neurologic decline are reported to range from 1.4% to 1.5%, with an annual mortality rate of 0%; therefore, treatment is still recommended in appropriately selected patients [66]. Endovascular treatment options for AVFs are like those of CCF and primarily consist of embolization with onyx, n-butyl cyanoacrylate, or coils, with the primary goal of disconnecting the fistulous point. One meta-analysis demonstrated that the use of onyx for transarterial embolization of intracranial dural fistulas was safe and effective, with low recurrence rates at midterm follow-up; however, the long-term risk was not assessed by this study [8].

Several recent studies have enhanced our knowledge of specific subtypes of intracranial AVFs. Lim et al. (2023) performed a systematic review of non-galenic pial AVFs (NGPAVFs), a rare pathology constituting only 1.6–4.8% of all cerebrovascular malformations [67]. Utilizing a cohort of 242 patients over 86 studies, the authors further characterized the clinical course and outcomes of this clinical entity: headache was the most common initial symptom (42.6%). Hemorrhage occurred with significantly higher frequency in adults, but congestive heart failure was more common in children. Finally, they found no difference in rates of complete NGPAVF obliteration among surgical, endovascular, or combination therapy (86.8%, 85.2%, and 88.5%, respectively), and 24.4% experienced some sort of complication with an overall mortality rate of 3.3%. A meta-analysis of treatment for ethmoidal dural AVFs found that surgical intervention was superior to endovascular treatment for complete obliteration rate but did not find any significant differences between postoperative transient ischemic attack, stroke, new-onset seizure, or ICH [68].

## 8. Other Extracranial Arteriovenous Fistulas

Extracranial AVF are rare lesions that may result from traumatic injury to the vessels in the neck, most commonly after penetrating neck injuries [69,70]. They typically arise from the vertebral artery, due to its close proximity to vertebral veins and the epidural venous plexus, but they can involve other vessels, such as the ECA branches, common carotid artery, and ICA [5,71,72,73]. Diagnosis of extracranial AVFs is challenging due to their often-delayed presentation and indolent course. Patients can present with a variety of symptoms depending on the location of the AVF, the degree of shunting, and subsequent intracranial venous congestion. Symptoms can range from mild manifestations such as dizziness, bruit, and pulsatile tinnitus to more severe ones like cervical myelopathy, subarachnoid hemorrhage, and cranial nerve deficits [5,70,71,72,73].

Noninvasive diagnostic imaging, such as color Doppler ultrasonography, MRA, and CTA, has taken an increasing role in the evaluation of these patients. CTA is particularly useful in the screening of penetrating trauma due to its ability to evaluate the entire cerebrovascular circulation from the aortic arch to the intracranial vessels [4,74,75]. However, DSA remains the most commonly used method for diagnosis and treatment as it enables the evaluation of arterial feeders, fistulous points, collateral circulation, and venous drainage. Furthermore, as the majority of extracranial AVFs arise from the surgically challenging vertebral artery, endovascular repair has emerged as a safe and feasible option, with reported occlusion rates of up to 89% [69]. Endovascular treatment options include detachable coils, stent placement, coil embolization, and liquid embolization, with the primary goal of obliterating the fistulous point [5,69,71,72,73]. We identified a single systematic review analyzing penetrating extracranial vertebral artery injuries in 462 patients who experienced injuries, including gunshot wounds, stab wounds, and other miscellaneous mechanisms [76]. The researchers identified several key features: the majority of cases present without neurological symptoms, although some present with exsanguinating hemorrhage; CTA should be considered first-line for diagnosis; and surgical ligation was the most common intervention, followed by angioembolization.

## 9. Epistaxis

Extensive skull base fractures can lead to cerebrovascular injuries, particularly in areas where vascular structures are fixed to the skull base [77]. Traumatic pseudoaneurysms resulting from these injuries may be asymptomatic or present with symptoms of a contained rupture, as in the case of CCF, or severe epistaxis. The incidence of intractable post-traumatic epistaxis ranges from 1 to 11% [78,79,80]. The most commonly involved vessel is the ECA, specifically the internal maxillary branch, followed by the cervical, petrous, and cavernous segments of the ICA [27,28].

Although magnetic resonance imaging and CTA are often the initial diagnostic tools, angiographic imaging remains the gold standard [28,78]. CTA has an overall sensitivity of 80% and may also be useful in guiding super-selective angiography necessary to demonstrate the lesion [5,6].

Severe craniofacial injury combined with intractable oronasal epistaxis is potentially fatal, with mortality rates reported between 30% and 50% [81,82]. The presentation may be delayed anywhere from three days to six months, with most cases presenting within the first three weeks [77]. Nasal packing or tamponade with balloon catheters can be effective in controlling mild to moderate bleeding; however, the effectiveness is reduced with severe bleeding [78,83]. If conservative techniques fail to achieve hemostasis, surgical therapy is often necessitated.

Interventional approaches to traumatic aneurysms have evolved from Hunterian parent vessel ligation to a largely endovascular paradigm. Success has been reported with the use of detachable coils, detachable balloons, covered stents, and flow-diverting stents [77]. Packing of the pseudoaneurysm with coils is feasible when the aneurysm neck is narrow, but it carries the risk of rupture in the acute phase [28,29]. If the aneurysm neck is narrow or the vessel is dysplastic, then balloon reconstruction, stent placement, or flow diversion is often preferred. Covered stents have demonstrated good efficacy in managing these lesions; however, they carry the risk of occluding potentially important branch vessels. For this reason, the use of flow-diverting stents is gaining popularity [30,31]. Nevertheless, the risk of bleeding in trauma patients from dual antiplatelet therapy necessitated by endovascular stent placement must be weighed against the risk of complications from an untreated pseudoaneurysm on an individualized basis. Parent artery occlusion is an additional management option that must be approached judiciously due to potential grave complications. When the lesion involves the ICA, a BTO is necessary to demonstrate adequate collateral circulation. Even after a successful BTO, up to 22% of patients may still develop complications following therapeutic vessel occlusion [84,85].

## 10. Cerebral Venous Sinus Thrombosis

Cerebral venous sinus thrombosis (CVST), with a reported incidence of up to 7%, is an entity that is frequently associated with head trauma and TBI [86]. Clinical features of CVST vary and can include headaches, seizures, and focal neurologic deficits. However, CVST can lead to significantly higher morbidity, particularly if involving the posterior third of the superior sagittal sinus or dominant transverse or sigmoid sinus. In these cases, venous infarct may result in catastrophic hemorrhage, elevation in intracranial pressure, coma, and death [87]. Poor prognostic indicators for CVST include coma at presentation, delineated infarction, and deep venous involvement [5,88]. CVST is commonly associated with disorders that cause systemic venous thrombosis, a diverse group of disorders that all induce a hypercoagulable state, which includes sickle cell anemia, polycythemia vera, essential thrombocythemia, pregnancy, and malignancies such as adenocarcinoma and leukemia [89]. This uncommon clinical entity has also gained increased attention recently due to its association with rare adverse events caused by COVID-19, as well as its vaccine [90,91,92]. In the trauma setting, CVST is frequently related to skull fracture, and the location of the fracture often determines which sinus is affected [93]. They may also occur secondary to external compression from an underlying hematoma [94].

Noninvasive imaging, such as CT venography and MR venography, is adequate for establishing a diagnosis, with sensitivities and specificities reported up to 95% [94,95,96]. A study by Hersh et al. found that of 113 patients who underwent formal venous imaging, 34% had sinus thrombosis, 17% of which demonstrated external compression by an extra-axial hemorrhage [86]. Interestingly, contrast-enhanced T1-weighted MRI is considered the most accurate for detecting dural venous sinus and CVST [97]. DSA is typically reserved for patients for whom endovascular therapy is being planned [84]. Patients with skull fractures or extra-axial hemorrhages near or overlying cerebral venous sinuses, or patients with a high suspicion of sinus injury, should be screened for CVST.

Treatment of CVST involves aggressive rehydration and anticoagulation, which often poses a challenge in patients with recent trauma. In patients with traumatic CVST treated with anticoagulation, rates of worsening intracranial hemorrhage have been reported to be up to 14%, with a 4.5% mortality rate [74]. Unfortunately, there are no clear guidelines specifying the appropriate time to initiate anticoagulation, although studies suggest that commencing anticoagulation 48–72 h after the initial injury is associated with fewer deleterious effects [98]. The optimal treatment duration is also unclear, but 3–6 months is generally accepted as an appropriate treatment course [84].

Patients who fail to improve or who have worsening neurologic deficits despite medical therapy may be considered for chemical or mechanical thrombolysis. Endovascular strategies include intrasinus thrombolysis, balloon angioplasty, thrombectomy, and sinus stent placement [99,100]. These techniques may be used in isolation or in combination. One systematic review demonstrated a 69% radiographic resolution rate with mechanical thrombectomy in medically refractory cases, with a 14.3% mortality rate reported during the follow-up period [101]. Additionally, recent reports have shown an improvement in intracranial pressure following mechanical thrombectomy and stent placement of dural sinuses [102].

## 11. Subdural Hematoma

Traumatic chronic SDH represents one of the most pervasive neurosurgical conditions, with an estimated annual incidence of 58.1–80.1 per 100,000 among individuals older than 65 years [103,104,105]. With the aging population, the incidence of chronic SDH is expected to double within the next 20 years [106]. Furthermore, 11–28% of patients who undergo surgical evacuation of a chronic SDH will experience recurrence [106,107,108]. The classical theory that SDH results from traumatic disruption of bridging subdural veins has been widely contested in recent literature. Emerging evidence instead suggests that the pathophysiology of chronic SDH is the result of local inflammation, causing hyperfibrinolysis of the clot and production of angiogenic factors that promote neovascularization and bleeding from fragile capillaries [109,110,111,112,113]. Delineation of these pathophysiologic mechanisms has facilitated the emergence of new and less invasive therapies.

While surgical evacuation remains the gold standard for alleviating mass effects from the hematoma, this does not treat the underlying neovascularization and inflammation, often allowing for the possibility of recurrence. As such, endovascular therapies are emerging as a viable treatment option. These therapies are based on the notion that embolization of middle meningeal artery (MMA) branches that supply the dura will inhibit blood influx into pathologic structures and control bleeding from the chronic SDH membrane, promoting spontaneous hematoma resolution.

Embolization may be performed with microparticles, onyx, n-butyl cyanoacrylate, or coils. A recent systematic review found that polyvinyl alcohol (PVA) was the most used embolic agent; however, the efficacy and rate of complications were not affected by the choice of embolisate [114]. Before embolization, selective angiography of the MMA is performed to target the branches of interest and avoid potentially dangerous collaterals (Figure 4). When flow stasis is observed in the MMA, the procedure is concluded.

In the largest reported series, 72 patients underwent MMA embolization, which was compared to 469 patients who underwent conventional treatment. In this study, all asymptomatic patients who underwent MMA embolization achieved spontaneous hematoma resolution. Of the 45 symptomatic patients, only one had hematoma re-accumulation. Importantly, the treatment failure rate was lower in the embolization group (1.4%) compared to the conventional treatment group (27.5%), and treatment complication rates were not different between the two groups [115].

A recent meta-analysis demonstrated a composite recurrence rate of 2.1% and 3.6% across double-arm and single-arm studies, respectively. These results were far lower than those reported for open surgical hematoma evacuation [116]. Functional outcomes, as assessed by the modified Rankin Scale (mRS), were not found to differ between embolization and conventional treatment groups, although this variable was not reported in select large studies. In the available literature, good outcome (mRS of 0–2) has been reported in up to 85% of patients [117].

While successful outcomes have been reported for embolization both before and after surgical evacuation, or as a stand-alone procedure, the ideal timing for MMA embolization remains to be determined. For patients with symptomatic or significant mass effects, surgical hematoma evacuation may still be necessary as the primary procedure, with MMA embolization serving as an adjunct procedure to prevent recurrence. As this is still a developing technique, further studies need to be performed to determine which patients will most benefit from MMA embolization.

## 12. Limitations

In addition to being a major limitation of this review, the scarcity of literature and completed RCTs examining neuroendovascular techniques has significant implications for the interpretation and application of medical treatments for cerebrovascular injuries [17,18,19]. When a consensus is lacking on the efficacy of therapeutic options, it becomes challenging for healthcare professionals to make well-informed decisions about which treatments are most effective and safe. RCTs are considered the gold standard for clinical research because they involve randomly assigning participants to different treatment groups, reducing bias and providing more reliable evidence. Additionally, as traumatic injuries are rare in comparison, much of the literature on these disease processes stems from non-traumatic etiologies. Consequently, the treatment algorithms for traumatic injuries are extrapolated from the literature on non-traumatic injuries, which may not accurately reflect the optimal approach for traumatic injuries. This uncertainty, particularly in the acute care of cerebrovascular injuries, can potentially result in suboptimal patient outcomes [1,14]. These limitations highlight the need for more research in these areas to provide robust evidence and improve patient care. As neuroendovascular techniques continue to evolve rapidly, the efficacy of newer therapies remains to be fully evaluated.

## 13. Future Directions

The rapid pace at which neuroendoscopic techniques emerge and evolve holds immense promise for expanding access to life-saving care and improving the treatment of cerebrovascular pathology, including craniocervical trauma. One exciting research opportunity is the potential for long-distance endovascular interventions using robotic endovascular surgery. This could provide access to life-saving care for those in remote areas, such as rural communities [92]. Additionally, the use of robotics in endovascular interventions has been shown to be more precise, reduce operational hazards for surgeons, and provide improved and more efficient outcomes for patients [92]. Attractive to both patients and neuroendovascular surgeons, the use of robotics targets one of the few drawbacks of the use of interventional radiology (IR), which is exposure to ionizing radiation [118].

The integration of artificial intelligence (AI) and machine learning into these technologies is another exciting frontier, with the potential to provide real-time feedback and surgical guidance, monitor organ movements, and enhance the prediction of postoperative outcomes. One recent narrative review of the use of AI in the clinical setting showed that, though quite preliminary, results are encouraging, and this technology is expected to have a substantial impact on the future of surgery and surgical training [119]. Several robotic systems for use in endovascular IR have been developed in the last two decades, including Sensei, Magellan, CorPath GRX, Amigo, Niobe RS, and R-ONE. In particular, the CorPath GRX has applications for use in neurovascular interventions, and several studies have been conducted and are currently active in further studying this system [9,91,120]. AI and other machine learning modalities can provide exceptional benefits to the surgical experience, such as real-time feedback and surgical guidance through the use of augmented reality combined with tension sensors in robotic arms to monitor organ movements, and enhanced prediction of postoperative outcomes. Robotic neuroendovascular surgery combined with AI and machine learning can help bring about true precision surgery and increase the overall quality of surgical interventions.

The rapid developments in neuroendovascular techniques bring forth both opportunities and challenges. One significant challenge is the imperative to thoroughly assess the safety and effectiveness of these emerging therapies through randomized controlled trials (RCTs) [17,18]. The pace of advancement may outstrip the research evaluating their efficacy, leading to delays in establishing algorithms with proven optimal treatments, potentially compromising patient outcomes. Concurrently, the adaptability and training of healthcare professionals in utilizing these new technologies pose another vital challenge that needs careful consideration [9,91]. Nevertheless, these challenges underscore the importance of ongoing research to rigorously evaluate the safety and effectiveness of these emerging therapies, while also providing comprehensive training for healthcare professionals to adeptly leverage these innovations.

## 14. Conclusions

The field of neuroendovascular techniques has evolved tremendously, adding to the armamentarium of neurosurgeons and trauma teams in the care of patients with traumatic head and neck injuries. Although improvements in noninvasive imaging modalities such as CTA and MRA have allowed for their increasing use as screening modalities, DSA remains the gold standard for diagnosing vascular injury as it allows for the safe and effective treatment of patients with carotid and vertebral dissections, pseudoaneurysms, various arteriovenous fistulas, and even subdural hematomas. As neuroendovascular teams become more prevalent and available for rapid interventions, endovascular techniques may take an expanded role in the management of acute trauma patients. However, further research and clinical trials, which can help inform future treatment guidelines which are lacking, are needed to fully delineate the role of neuroendovascular techniques in the management of traumatic head and neck injuries.

## Figures and Tables

**Figure 1 biomedicines-11-02409-f001:**
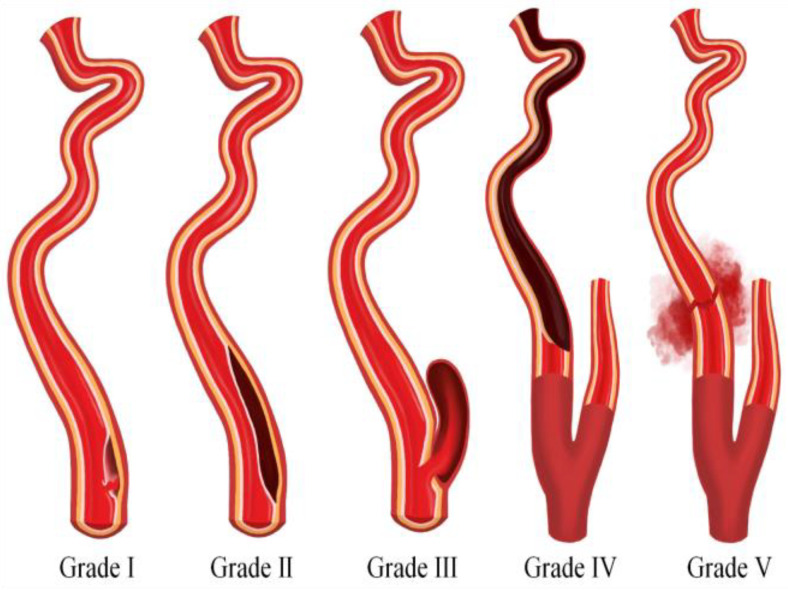
An illustration depicting the five grades of blunt cerebrovascular injury.

**Figure 2 biomedicines-11-02409-f002:**
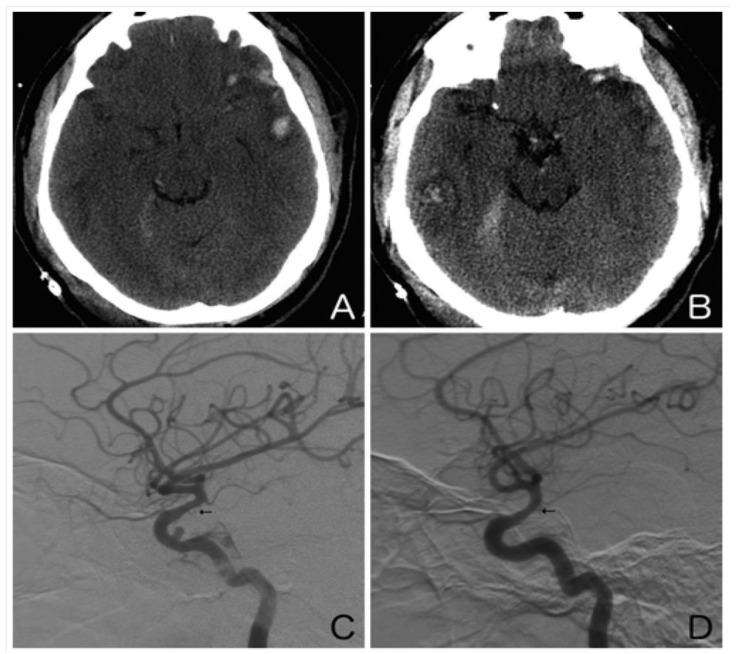
Illustrative case. A 25-year-old man presented after an assault to the head. An axial non-contrast computed tomography scan of the head showed a left temporal contusion with adjacent subarachnoid hemorrhage (**A**) and a right temporal contusion and tentorial subdural hematoma (**B**). Right (**C**) and left (**D**) internal carotid artery digital subtraction angiograms (lateral view) obtained on day 10 showed moderate vasospasm of both supraclinoid internal carotid arteries (arrows). Note the traumatic pseudoaneurysm of the cavernous right internal carotid artery.

**Figure 3 biomedicines-11-02409-f003:**
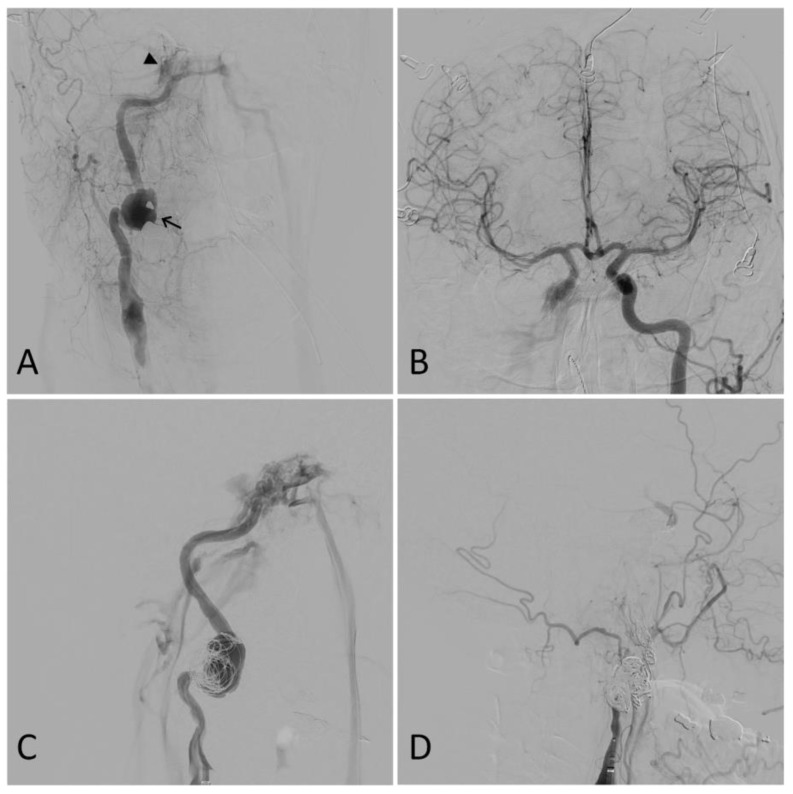
Illustrative case. A 50-year-old male presented after a motor vehicle accident. A computed tomography angiogram revealed a right internal carotid artery (ICA) pseudoaneurysm with associated stenosis. A digital subtraction angiogram showed a 27 mm right ICA dissecting pseudoaneurysm (arrow) and a right carotid-cavernous fistula (CCF) (arrowhead) (**A**). Furthermore, the right anterior circulation was noted to fill completely through the left ICA (**B**). Partial coil embolization was performed at this time to protect the dome of the pseudoaneurysm (**C**). A repeat angiogram two weeks later showed the right ICA was dissected throughout its entire course and ended in a false lumen in the supraclinoid segment with no antegrade flow or intracranial filling. The CCF, pseudoaneurysm, and dissected ICA were then coiled. A final right common carotid angiogram showed no filling of the ICA, pseudoaneurysm, or CCF (**D**).

**Figure 4 biomedicines-11-02409-f004:**
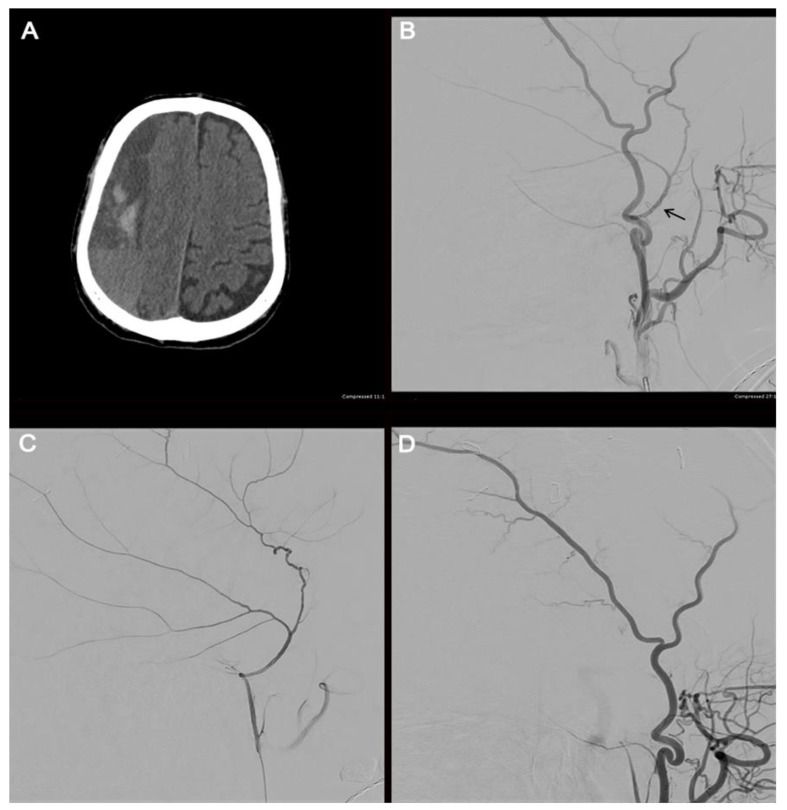
Illustrative case. The radiographic course of an 82-year-old male who presented with a large, septated, chronic right-sided SDH (**A**). The patient underwent a cerebral angiogram with embolization of the right middle meningeal artery using 50–150 µm Embospheres. An external carotid artery angiogram (**B**) shows the filling of the middle meningeal artery (arrow). A selective middle meningeal artery angiogram was then performed (**C**). A post-embolization angiogram of the external carotid artery shows no filling of the middle meningeal artery (**D**).

**Table 1 biomedicines-11-02409-t001:** BCVI Management.

Grade	Description	Management	Outcomes	Sources
I–II	I—intimal irregularity; <25% vessel stenosis II—dissection; >25% vessel stenosis	- Antithrombotic agents (e.g., aspirin); - Stenting is no longer recommended as an adjunct.	- High rates of resolution; - 50% of untreated healed at follow-up; - ~1% secondary stroke risk.	[14,22,23]
III	Pseudoaneurysm	- Antithrombotic agents; - Interventions recommended for pseudoaneurysms >1.0 to 1.5 cm; - Coil embolization +/− stent placement (e.g., covered stents, flow-diverters).	- Moderate success rate; - >50% remained the same or enlarged at 6 months; - 7% secondary stroke risk.	[5,7,23,27,28,29,30,31]
IV	Occlusion	- AC/AP therapy; - Mechanical thrombolysis/thrombectomy; - Coil/liquid embolization.	- High acute mortality; - 82% unchanged on follow-up; - 7% secondary stroke risk.	[5,7,23,24,25]
V	Transection of the extracranial carotid or vertebral arteries with active extravasation	- Urgent hemorrhage control (i.e., direct pressure); - Surgical repair for accessible lesions; - Endovascular techniques (e.g., stent, BTO) for inaccessible injuries.	- High morbidity and mortality; - Nearly 100% secondary stroke risk.	[5,18]

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
