# Peer review of "Neuroendovascular Surgery Applications in Craniocervical Trauma"

_biomedicines, 2023, doi:10.3390/biomedicines11092409_

Round 1

Reviewer 1 Report

The aim of this narrative review is to provide an updated and comprehensive nonsystematic review of the role of endovascular treatments in the management of traumatic cerebrovascular injuries 

The topic is interesting. However, the review interval is too long. I would recommend to limit the review to more recent years.

Also, I would appreciate to have data summarized in tables. In particular, pros and cons of each technique for every disease should be represented in tables.

Reviewer 2 Report

Cerebrovascular injuries resulting from blunt or penetrating trauma to the head and neck often lead to local hemorrhage and stroke, presenting with a wide range of manifestations, including carotid or vertebral artery dissection, pseudoaneurysm, occlusion, transection, arteriovenous fistula, carotid cavernous fistula, epistaxis, venous sinus thrombosis, and subdural hematoma. A selective review of the literature from 1989 to 2020 was conducted to explore various neuroendovascular surgical techniques for craniocervical trauma was performed. A PubMed search was performed using the terms: endovascular, trauma, dissection, blunt cerebrovascular injury, pseudoaneurysm, occlusion, transection, vasospasm, carotid-cavernous fistula, arteriovenous fistula, epistaxis, cerebral venous sinus thrombosis, subdural hematoma, and middle meningeal artery embolization. An increasing array of neuroendovascular procedures are currently available to treat these traumatic injuries. Coils, liquid embolics (onyx or n-butyl cyanoacrylate), and polyvinyl alcohol particles can be used to embolize lesions, while stents, mechanical thrombectomy employing stent-retrievers or aspiration catheters, and balloon occlusion tests and supraselective angiography offer additional treatment options based on the specific case. Neuroendovascular techniques prove valuable when surgical options are limited, although comparative data with surgical techniques in trauma cases is limited. Further research is needed to assess the efficacy and outcomes associated with these inter- ventions. However, there are some problems. And I have some suggestions for the content and structure of the article.

1)   The article may not sufficiently explain the advantages and disadvantages of each treatment method when discussing them. For instance, the article mentions endovascular techniques but does not delve into the potential risks or limitations of this method. This could make it difficult for readers to fully understand the applicability and effectiveness of each treatment.

2)   When discussing research findings, the article may not provide enough data or charts to support its points. For example, the article mentions the success rate of treatments but does not provide specific data or studies to back up this claim. This could make it difficult for readers to understand and evaluate the actual effects of the treatments.

3)   In discussing the disease process, the article may not sufficiently explain the pathophysiological mechanisms of the disease. For instance, the article mentions traumatic injuries but does not elaborate on how these injuries occur and how they impact the patient's health. This could make it difficult for readers to fully understand the progression and impact of the disease.

4)   When discussing the limitations of the research, the article may not sufficiently explain how these limitations affect the interpretation and application of the research findings. For example, the article mentions the scarcity of literature and randomized controlled trials but does not elaborate on how this affects the evaluation of the effectiveness of treatment plans. This could make it difficult for readers to understand and evaluate the reliability and applicability of the research.

5)   The article may not sufficiently discuss future research directions. For instance, the article mentions the rapid development of neuroendoscopic techniques but does not elaborate on the new research opportunities or challenges this may bring. This could make it difficult for readers to understand and evaluate the future trends in research.

6)   When discussing treatment plans, the article may not sufficiently explain how these plans adapt to different patients and disease states. For instance, the article mentions observation and medical therapy but does not elaborate on how these plans adapt to different patients and disease states. This could make it difficult for readers to understand and evaluate the applicability and effectiveness of the treatment plans.

7)   In discussing the diagnosis and assessment of the disease, the article may not sufficiently explain the importance and challenges of these processes. For instance, the article mentions standard ocular tonometry and color Doppler imaging but does not elaborate on the advantages and disadvantages of these methods, as well as the challenges that may be encountered in practical applications. This could make it difficult for readers to understand and evaluate the importance and challenges of the diagnosis and assessment processes.

8)   More solutions published 2021-2023 should be discussed.

9)   More quantitative analysis should be given.

10)     More technical details should be discussed.

NA

Round 2

Reviewer 1 Report

Thank you to the Authors for their efforts in the attempt to ameliorate their paper.

In table I would like to see percentage of success rates.

Reviewer 2 Report

All my concerns have been addressed. I recommend this paper for publication.

NA

Author Response

Thank you again for your feedback